# Please Don’t Compliment Me! Fear of Positive Evaluation and Emotion Regulation—Implications for Adolescents’ Social Anxiety

**DOI:** 10.3390/jcm11205979

**Published:** 2022-10-11

**Authors:** Achilleas Tsarpalis-Fragkoulidis, Rahel Lea van Eickels, Martina Zemp

**Affiliations:** Department of Clinical and Health Psychology, University of Vienna, 1010 Vienna, Austria

**Keywords:** fear of positive evaluation, fear of negative evaluation, social anxiety, emotion regulation, positivity impairment, adolescence

## Abstract

In recent years, fear of positive evaluation has emerged as one of the key aspects of social anxiety, alongside fear of negative evaluation. Fears of evaluation intensify during adolescence, a time when individuals are expected to navigate new, emotionally challenging situations. The purpose of this study was to examine the associations between social anxiety, fear of positive and negative evaluation, and three emotion regulation strategies relevant to social anxiety, i.e., suppression, acceptance, and rumination. To this end, data were collected from 647 adolescents via an online survey and analyzed using structural equation modeling. We found that fear of negative evaluation was significantly related to rumination, whereas fear of positive evaluation was significantly and negatively related to acceptance. We further found an indirect effect of social anxiety on suppression via fear of positive evaluation and acceptance in a serial mediation and an indirect effect of social anxiety on rumination via fear of negative evaluation. Not only do fears of positive and negative evaluation appear to be distinct constructs, but they are also differentially associated with three emotion regulation strategies pertinent to social anxiety. Fear of evaluation and its associations with emotion regulation deficits might hinder the therapeutic process by acting as a deterrent to positive reinforcement or potentially impeding the development of a successful therapeutic alliance.

## 1. Introduction

Social anxiety disorder (SAD) is one of the most common anxiety disorders in late adolescence, with prevalent rates ranging from 5% to 10% and a lifetime prevalence of around 12% [1,2,3]. Often first occurring in childhood or early adolescence, SAD has relatively low remission rates [4], usually persists into adulthood [5], and frequently leads to the development of other comorbid disorders, most notably, other anxiety, depressive, and substance use disorders [6,7,8]. Given that adolescence is a developmental period characterized by cognitive maturation, increasing affective reactivity, growing autonomy, and new socialization pressures [9], fears of evaluation (e.g., being afraid of being negatively evaluated by others, fear of being teased) appear to intensify during this period, while other fears, such as the fear of physical punishment, decrease [10]. These far-reaching changes place strain on the emerging adolescents and require a new set of skills to regulate the emotions that accompany these novel, frequent, and intense emotional challenges [11]. Although the increase in evaluation fears at this developmental stage can be regarded as a natural result of socio-cognitive maturation, it may also constitute a risk factor for the development of social anxiety at both subclinical levels as well as the clinical presentation of SAD [12]. Similarly, the inability to navigate these new emotional challenges in an appropriate and adaptive manner can be seen as an obstacle to successful socio-emotional adjustment, posing additional risks for the development of a variety of psychological problems, including social anxiety [11,13].

### 1.1. Fear of Evaluation

Fear of negative evaluation (FNE) has long been considered to be the key aspect of social anxiety [14,15]. In the DSM-5 and ICD-11, it is listed as a diagnostic criterion of SAD, in that affected individuals are fearful and avoidant of social situations in which they could be observed and evaluated by others [16,17]. FNE has consequently been included in multiple theoretical models of social anxiety, such as the cognitive model of social phobia [14] and the cognitive–behavioral model of SAD [15]. According to these models, socially anxious individuals who find themselves in social or performance situations shift their attention inwards, monitoring the self for any display of stress symptoms or signs of inadequacy, and outwards, searching for possible signs of negative evaluation, while positive cues are either not attended to or misinterpreted [18]. Stress symptoms, such as elevated heart rate, sweating, or blushing, manifest before and during these situations and, combined with the aforementioned attentional biases, create a vicious cycle, often leading to a steady increase in anxiety levels [6,16]. As a consequence, affected individuals report impairments in a plethora of life domains, such as work and studies, social relationships, leisure, and family life [19], which, in turn, amplify the symptoms in the long term. These underlying processes stress the importance of cognitive biases in the context of social anxiety and clearly indicate how FNE may lead to dysfunctional affective, cognitive, and behavioral responses to social situations [20].

In the past decade, however, researchers have argued that it is fear of evaluation in general, i.e., fears of negative and positive evaluation, that plays a vital role in social anxiety [21,22]. Fear of positive evaluation (FPE), which is defined as “feelings of apprehension about others’ positive evaluations of oneself and distress over these evaluations” [23] (p. 433), has been shown to be associated with a number of characteristics of social anxiety, including, but not limited to, submissive behaviors, increased negative affect and decreased positive affect in social interactions, and discomfort when receiving positive feedback [21,24]. This lack of positive affect and positive interactions, also referred to as positivity impairment, has gradually emerged as one of the central features of SAD [25,26,27]. Strikingly, FPE appears to contribute uniquely to social anxiety, above and beyond FNE, suggesting that both types of fear of evaluation, albeit correlated, are distinct constructs and that FPE might be the driving force for the positivity impairment observed in SAD [20,28]. FPE has also been successfully assessed in adolescent samples, where similar associations with social anxiety, avoidance, and safety-seeking behaviors have been found [28].

In order to understand the role of evaluation fears in the context of social anxiety, it is worth considering the evolutionary benefits of these fears. According to the psycho-evolutionary model of SAD [29], socially anxious individuals perceive themselves to be placed low in a hierarchically structured social environment. Driven by the attempt to avoid conflict with higher-ranking individuals, on the one hand, and general exclusion from the group, on the other, FNE and FPE are posited to play the role of regulatory forces that reduce the likelihood of such upward or downward movements in the social hierarchy. FNE may serve the function of protection from group exclusion, while FPE may protect against performing “too well” and, as a result, coming into conflict with the higher-ups [20]. However, when these processes become excessive, they appear to lead to a variety of negative outcomes, contributing to the development and maintenance of social anxiety [28,30].

Although FPE has shown promise as an approach to social anxiety research that could expand our understanding of its phenomenology, there are still many unanswered questions [20]. Specifically, little headway has been made in examining the associations between FPE and emotion regulation and their meaning for social anxiety.

### 1.2. Emotion Regulation

Emotion regulation is defined as “the processes through which individuals influence which emotions they have, when they have them, and how they experience and express these emotions” [31] (p. 275). According to this definition, emotion regulation deficits can manifest at different stages of the emotion generative process, with a broad distinction being made between antecedent-focused and response-focused strategies. Antecedent-focused strategies can occur before an emotion has been elicited, for instance, by shifting attention away from or reframing the meaning of a potentially emotional situation (i.e., attentional deployment and cognitive change), whereas response-focused strategies occur after an emotion has been triggered and aim to modulate the experience of said emotion (i.e., response modulation) [32]. Dysfunctional emotion regulation constitutes a trans-diagnostic process fundamental to and present in many different mental disorders, including SAD [33,34]. More specifically, social anxiety, at both clinical and subclinical levels, has been linked to the excessive use of various emotion regulation strategies that are generally deemed maladaptive, such as suppression [35,36] and rumination [37], accompanied by difficulties in accepting negative emotions [38]. Suppression and acceptance are seen as response-focused emotion regulation strategies since they mainly take place after an emotion has been elicited [39]. Rumination, on the other hand, is characterized by a perseverative focus on mainly negative emotional experiences, their causes, and potential consequences, thereby reflecting difficulty in redirecting attentional resources away from negative thoughts [40].

First, suppression can be defined as the voluntary inhibition of verbal and behavioral expressions of emotions [32]. The often maladaptive nature of emotion suppression lies in the fact that it usually does not achieve the goal for which it is used; on the contrary, negative emotions tend to become more intense when they are suppressed, leading to an increase in the subjective experience of those emotions [35]. In contrast to negative emotions, suppressing positive emotions does indeed dampen the intensity of those emotions and is linked to fewer positive emotions and experiences [41,42]. Given this counterproductive effect, it becomes clear how the excessive use of suppression in the context of social anxiety is strongly related to a variety of negative outcomes, such as lower life satisfaction, dysfunctional beliefs about the nature of emotions, and a general fear of emotional experiences [43,44]. Furthermore, suppression is thought to be related to the general tendency to avoid being in the spotlight because expressions of emotion might attract attention and, consequently, raise the chance of being evaluated by others [35,36,45,46].

Second, rumination, which represents an aspect of what is known as repetitive negative thinking in the literature, refers to thoughts that are “repetitive, intrusive, difficult to disengage from, perceived as unproductive, and capturing mental capacity” [47] (p. 441). In the case of social anxiety, these ruminative processes manifest mainly as post-event processing, a phenomenon that refers to the tendency to brood over past social events by selectively recalling negative information and negatively evaluating one’s own performance in a particular social situation [37,48]. Even in the case of events that include positive feedback, socially anxious individuals have been shown to focus on and brood over the negative aspects of these events [37]. This negative interpretation bias has also been linked to attempts to dampen positive emotions that might arise from positive social events. This suggests that as potentially disconfirmatory positive information is presented to socially anxious individuals, the positive emotional response it might elicit is dampened, thus maintaining the initial negative interpretations [49]. Findings on differences between socially anxious individuals vs. non-socially anxious controls regarding positive rumination, however, are not that clear; socially anxious individuals appear to focus more on the negative aspects of social events but not less on the positive aspects of them [50]. Although referred to as *post*-event processing, ruminative processes may also pertain to future events, in that socially anxious individuals might brood over what they perceive as past social “failures” before entering a new social situation, further exacerbating their anticipatory anxiety [51]. Ruminative thoughts can ultimately reinforce negative self-images, thus perpetuating the cycle of social anxiety [52].

Third, a fundamental regulatory problem in social anxiety lies in an unwillingness to accept negative emotions [38,43,53,54]. This is in line with findings on experiential avoidance in social anxiety, suggesting that socially anxious individuals are reluctant to face naturally occurring negative affect and try to avoid them through maladaptive strategies, such as attempts to conceal the emotional experiences [35,55]. Regarding positive emotions, socially anxious individuals have been theorized to have difficulties accepting them as well. For example, a general tendency to dampen positive emotions instead of accepting them as they are has been linked to social anxiety [56]. In addition, as mentioned above, individuals that have difficulty rectifying negative interpretations when presented with positive information tend to respond to positive emotions by attenuating their intensity and duration [49]. It thus becomes clear how difficulties with accepting emotional experiences have been linked to rumination [57]: Given how negative interpretation biases place the focus on the negative aspects of a situation, these interpretations are difficult to re-adjust when positive information is presented, and positive emotions that might be triggered in these situations are immediately dampened.

Although most of the aforementioned studies have been conducted with adult samples, the majority of these findings have also been replicated with children and adolescents. In fact, associations between adolescents’ social anxiety and acceptance [58,59,60], suppression [61,62], and rumination [61,63,64,65,66] have been found in both clinical and community youth samples.

### 1.3. Do Evaluation Fears Mediate the Link between Social Anxiety and Emotion Regulation?

In sum, FPE appears to be strongly and uniquely associated with social anxiety, often above and beyond FNE [20,28]. Most notably, it is thought to drive the observed positivity impairment in the disorder, in that it is closely connected with low positive affect, disqualification of positive social events, and less perceived accuracy of positive feedback, whereas FNE appears to be more specifically related to excessive negativity [24,67]. Furthermore, socially anxious individuals seem to be less accepting of their emotions [43], suppress them more often [46], and spend more time ruminating over past or future events [37,51]. It is thought that suppression might occur out of fear of being in the spotlight and consequently evaluated; ruminative processes place the focus on past social situations and perceived negative evaluations of the self, whereas non-acceptance might play a role in the disqualification of positive social events and the concomitant positive emotions as well as in the avoidance of negative emotional experiences. These regulatory difficulties appear to contribute to this positivity impairment as well, since these frequently employed strategies intensify negative emotions and dampen positive emotions, increase negative cognitions, and are generally associated with less positive events in everyday life [35,42,68,69].

That said, to our knowledge, no study has yet investigated the associations between fear of evaluation (i.e., both FNE and FPE) and the three emotion regulation strategies (i.e., suppression, rumination, and acceptance) in the context of social anxiety in adolescents. Moreover, it has not been examined, so far, whether evaluation fears explain the associations between social anxiety and emotion regulation deficits. As mentioned above, FNE and FPE are assumed to play the role of regulatory forces with respect to upward and downward movements in the social hierarchy [20,29] and, by proxy, emotional-expressive behavior. Therefore, it is plausible that the two types of fear of evaluation will mediate the link between social anxiety and the various emotion regulation strategies among adolescents.

### 1.4. The Current Study

The goal of this study was to examine the associations between the fears of negative and positive evaluation (i.e., FNE and FPE) and three emotion regulation strategies (i.e., suppression, rumination, and acceptance) and to investigate their role in social anxiety in adolescents. When examining the associations between fears of evaluation and emotion regulation strategies, we consider it essential to investigate them simultaneously, that is, to control for one when examining the other, in order to test their unique contributions and avoid associations that may result from nonspecific evaluative fears. First, we expected social anxiety to be positively associated with both FPE and FNE (H1). Furthermore, we assumed that FNE would be positively associated with suppression and rumination and negatively associated with acceptance after controlling for FPE (H2). With respect to FPE, we expected it to be associated with acceptance and suppression but not rumination after controlling for FNE (H3). We further hypothesized that the two types of fear of evaluation would be the mechanisms that mediate the links between social anxiety and emotion regulation strategies differentially. More specifically, we expected FNE to mediate the association between social anxiety and suppression as well as rumination (H4a and H4b) and FPE to mediate the association between social anxiety and suppression (H5) but not rumination. Additionally, we predicted that both FNE and FPE will mediate the association between social anxiety and acceptance (H6). Given that difficulties in accepting negative emotions in the context of social anxiety can lead to attempts to hide emotional experiences, we also assumed that acceptance will predict suppression in our model (H7). Since rumination is also negatively associated with acceptance, we hypothesized that acceptance would also be linked to rumination (H8). With these specifications, we expected the indirect effects of social anxiety on suppression and rumination to be mediated by FPE, FNE, and acceptance in the form of a double mediation (H9). We controlled for adolescents’ age, gender, and depressive symptoms in all analyses. Figure 1 illustrates our hypothesized model.

## 2. Materials and Methods

### 2.1. Participants

A convenience sample was pulled from the general German-speaking adolescent population (14–17 years) in Austria and Germany. Participants were mainly recruited via advertisements on social media, such as Facebook and Instagram. In total, 1049 participants started the survey, of whom 724 reached the last page. Participants were informed on the first page of the questionnaire that their data would not be used for the analysis if they stopped answering the questionnaire at any point. Therefore, only data from participants who reached the last page were utilized for data analysis. In addition, at the end of each page of the questionnaire, participants were prompted to answer all questions if any response was missing. If they chose to intentionally leave a question unanswered, they had the option to check a box and continue to the next page. This resulted in a final data set with virtually no missing data, except for one respondent who left an item unanswered (FPE4), evidently deliberately.

Of the 325 participants who did not reach the last page, 102 (31.4%) answered the sociodemographic questions and then dropped out afterwards. Strikingly, 77 participants (23.7%) quit the survey without completing the next page, where we had placed a CAPTCHA to avoid automated responses, which has recently emerged as a serious problem in online surveys with monetary incentives [70]. The remaining 146 participants (44.9%) dropped out during the remainder of the survey. We then conducted an attrition analysis to examine potential differences between those who completed the survey and those who did not. This revealed no significant differences in terms of sociodemographic variables (gender, country of residence, native language, school type, school level, psychotherapeutic treatment, lockdown, quarantine). These analyses have been made available online and can be retrieved from OSF (see the data availability statement).

After completing internal testing, we excluded the quickest and slowest 5% of the respondents (10% in total, i.e., 72 participants) from the final dataset. We deemed these participants to be non-serious respondents, given that they completed some of the questionnaire pages in as quickly as 20 s or as slowly as 10 min per page. Furthermore, we examined univariate outliers using boxplots and screened these participants’ response behavior for peculiar responses, which did not result in any exclusions. Additionally, we calculated Mahalanobis distances using our variables of interest to detect possible multivariate outliers, which, after examining the chi-squared distribution of the distances at *p* < 0.001, resulted in the exclusion of four additional participants [71]. Lastly, we found one participant whom we determined to be a non-serious responder, given that their responses were consistently at opposite extremes on unidimensional, straight-forward coded scales (e.g., 0, 9, 0, 9, 0, 9, etc. on the scale of FPE). Thus, the final sample consisted of 647 participants (85.6% female, 10% male, 4.3% other) with a mean age of 16.21 (*SD* = 0.95) years. Further sociodemographic characteristics are shown in Table 1.

### 2.2. Procedures

This study is part of a larger longitudinal project that aims to examine the prospective associations and underlying mechanisms of fear of evaluation, emotion regulation, and social anxiety in adolescents. This study was preregistered on 10 March 2022. Although the preregistration was finalized after data collection had begun, no data had been screened or reviewed yet. The study’s preregistration can be retrieved from OSF at osf.io/fgeb3 (accessed on 4 October 2022).

Only data from the first wave of surveys were used for this report, which was collected by means of an anonymous online survey on the SoSci Survey platform [72] from 15 February 2022 to 25 June 2022. In order to increase the participation rate, participants had the opportunity to take part in a draw for 10 vouchers worth 10 euros each for an online store of their choice (e.g., video games, fair trade stores). Participation was voluntary and could be terminated at any time by closing the browser window. The adolescents’ informed consent was required to start the survey.

### 2.3. Measures

#### 2.3.1. Social Anxiety

Social anxiety was measured using the Social Phobia Inventory (SPIN) [73,74]. The SPIN consists of 17 items related to three aspects of social anxiety, i.e., fear of social situations, avoidance of social situations, and physiological symptoms of anxiety. Participants are asked to indicate on a 5-point rating scale how anxious they have felt about their behavior in social situations in the past two weeks. Because the discriminatory power of some items (e.g., “I avoid going to parties”) could potentially be affected by the COVID-19 pandemic, we added a disclaimer as part of the instructions and urged participants to think of periods when the restrictions were less severe and changed the time reference from two weeks to three months. This amendment was approved by the publisher of the instrument. The statements are rated on a scale ranging from 0 = *not at all* to 4 = *extremely*, with a maximum sum score of 68 (higher scores reflect greater social anxiety). However, mean scores were used for our main analysis. This scale showed acceptable internal consistency (Cronbach’s α = 0.93).

#### 2.3.2. Fear of Evaluation

FPE was assessed using the Fear of Positive Evaluation Scale (FPES) [22,75]. The FPES consists of ten statements related to fear and discomfort when receiving positive attention (e.g., “I feel uneasy when I receive praise from authority figures”). These are rated based on a 10-point rating scale, from 0 = *not at all true* to 9 = *very true*, with higher scores indicating a higher fear of positive evaluation. Since two of these items displayed insufficient discriminatory power (<0.30) in previous validation studies, they were excluded from the analysis, meaning the final score was comprised of eight items, all loading on a single factor. The internal consistency of the scale was acceptable, with Cronbach’s α = 0.87.

FNE was measured with the Brief Fear of Negative Evaluation Scale (BFNE) [76,77]. The BFNE consists of twelve items related to concerns about being criticized or found inadequate by others (e.g., “I am afraid that others will not approve of me”). They are rated on a 5-point rating scale from 1 = *not at all characteristic of me* to 5 = *absolutely characteristic of me*, with higher scores reflecting a higher fear of negative evaluation. Similar to the FPES, these items loaded on a single factor and displayed acceptable internal consistency (Cronbach’s α = 0.95).

#### 2.3.3. Emotion Regulation

Suppression and acceptance were assessed with the Affective Style Questionnaire–Youth (ASQ–Y) [39,78]. This questionnaire consists of 20 items, 8 of which measure suppression/concealingt (e.g., “I often suppress my emotional reactions to things”) and 5 of which measure acceptance/tolerating (e.g., “It’s ok to feel negative emotions at times”). The statements refer to participants’ habitual responses to emotional experiences and are rated on a 5-point rating scale, ranging from 0 = *not true of me* at all to 4 = *extremely true of me*. Higher scores reflect higher suppression and higher acceptance, respectively. Both scales had satisfactory internal consistencies (Cronbach’s α = 0.84 for suppression and α = 0.84 for acceptance).

Rumination was assessed with the Perseverative Thinking Questionnaire (PTQ) [79]. The PTQ consists of 15 items distributed across three factors, i.e., the core characteristics of repetitive negative thinking (e.g., “The same thoughts keep going through my mind again and again”), the unproductiveness of repetitive negative thinking (e.g., “I think about many problems without solving any of them”), and repetitive negative thinking impairing mental capacity (e.g., “My thoughts take up all my attention”). Participants are asked to reflect on how they typically think about negative experiences and problems and to rate the statements on a 5-point rating scale, with 0 = *never* and 4 = *almost always*. Higher scores represent higher repetitive negative thinking. The three subscales displayed acceptable internal consistency (core characteristics of repetitive negative thinking: α = 0.90; unproductiveness of repetitive negative thinking: α = 0.76; repetitive negative thinking impairing mental capacity α = 0.83). The overall scale, which was used in the present analysis, as recommended by the scale’s creators, and hereinafter consistently denoted as rumination, had an internal consistency of α = 0.93 and, thereby, acceptable reliability.

#### 2.3.4. Control Variables

The depression module of the Patient Health Questionnaire (PHQ-9) [80,81] was used to measure depressive symptoms to be included as a covariate in the analyses. Depressive symptoms are likely to be a confounding factor in our analyses because prior research has found strong associations between depressive symptoms and social anxiety [82] as well as the three emotion regulation strategies examined in this study [33]. Participants were thereby asked to report how often they felt impaired by a series of depressive symptoms (e.g., “Little interest or pleasure in your activities”) in the past two weeks. The scale is comprised of nine items that are rated on a 4-point rating scale, from 0 = *not at all* to 3 = *almost every day*. Higher scores reflect higher depressive symptoms. The scale displayed an acceptable internal consistency of α = 0.87.

Additionally, we controlled for adolescents’ age and gender, given that previous studies have reported age and gender effects in terms of social anxiety [83] and emotion regulation [84]. Age was assessed as a continuous variable via free text-input. For gender, we created two dummy-coded variables (0 = female, 1 = male, 2 = other gender; reference category = female).

### 2.4. Statistical Analysis

We used IBM SPSS 27 Statistics for Windows Version 27 (IBM Corp, Armonk, NY, USA) [85] for descriptive statistics, outlier detection, reliability analyses, and assumption testing. For confirmatory factor analyses and structural equation modeling, we used Mplus 8.5 (Muthén & Muthén, Los Angeles, CA, USA) [86]. We conducted confirmatory factor analyses with our main questionnaires, i.e., SPIN, FPES, BFNE, ASQ–Y, PTQ, and PHQ-9, to examine factor loadings and calculate composite reliabilities. Following the suggestions of Maydeu-Olivares [87], we estimated these models using mean-variance-adjusted maximum likelihood estimation (MLMV), which has been shown to display accurate standard errors and outperform traditional maximum likelihood estimation in terms of goodness-of-fit testing in the presence of a few pieces of missing data. For the structural equation model, we used bootstrapping and requested bias-corrected confidence intervals. Model fit was assessed using the cut-offs for CFI, TLI, RMSEA, and SRMR, as provided by Hu and Bentler [88], aiming for at least acceptable values for all fit indices (CFI/TLI > 0.90; RMSEA/SRMR < 0.08).

## 3. Results

### 3.1. Descriptive Statistics

Means, standard deviations, and bivariate correlations of the study variables are shown in Table 2. All analyses were calculated using mean scores across all measures and will hereinafter be reported accordingly. Bivariate correlations revealed that our study variables were correlated in the expected direction, with both fears of evaluation exhibiting positive associations with suppression and rumination, and negative links to acceptance. Moreover, social anxiety was associated with both fears of evaluation and all emotion regulation strategies. An additional set of multiple linear regressions were performed to test for multivariate normality, homoscedasticity, linearity, residual independence, and multicollinearity. As is common in psychopathological measures with community or subclinical samples, we found that some of our variables were not normally distributed. We used mean-variance-adjusted maximum likelihood estimation (MLMV) and bootstrapping for all of our main analyses, which are both viable options for dealing with non-normality within SEM [89,90]. Our analyses revealed no other violations of the statistical requirements. These analyses have been made available online and can be retrieved from OSF (see the data availability statement).

### 3.2. Measurement Model

In the first step of our analysis, we conducted CFAs with all key measures of this study to examine factor loadings and composite reliabilities. For this purpose, we used the fixed factor method to estimate composite reliability (McDonald’s omega), which is indicated over Cronbach’s alpha when dealing with tau-congeneric measurement models [91]. All CFAs yielded acceptable model fits and reliability coefficients (see Table 3). Second, we calculated the means and variances of the study variables. Finally, we used the mean scores of the observed variables as single indicators of their respective latent variables, fixing factor loadings at 1 and error variances at
δ_x_ = VAR(X) × (1 − ω)(1)
to control for measurement error ([92,93], p. 139).

### 3.3. Structural Model

For our structural equation model, we conducted the analysis using bootstrapping with 5000 samples and examined the indirect effects using bias-corrected 95% confidence intervals. In terms of the structural relationships, we first regressed FPE and FNE on social anxiety. We then regressed the three emotion regulation strategies, i.e., acceptance, suppression, and rumination, on social anxiety and both fears of evaluation. This allowed us to examine the effect of each type of fear of evaluation on the three emotion regulation strategies while controlling for the other. We also regressed rumination and suppression on acceptance. Finally, we specified the indirect effects of social anxiety on all three emotion regulation strategies via FPE and FNE. FPE and FNE were additionally regressed on gender and age, while all emotion regulation variables were regressed on gender, age, and depression. The model fit was acceptable (CFI = 0.994, TLI = 0.972, RMSEA = 0.045 [0.020, 0.071], SRMR = 0.034).

Regarding our hypotheses, we found that social anxiety was positively associated with FNE (β = 0.709, *p* < 0.001) and FPE (β = 0.779, *p* < 0.001). In addition, FNE was positively associated with rumination (β = 0.330, *p* < 0.001) but not with acceptance or suppression. FPE was negatively associated with acceptance (β = −0.339, *p* < 0.001) but not with suppression or rumination. Acceptance was negatively linked to suppression (β = −0.502, *p* < 0.001), as was social anxiety (β = −0.234, *p* = 0.018). All direct effects are displayed in Table 4.

Concerning our mediation hypotheses, we found significant indirect effects of social anxiety on acceptance via FPE (single mediation; β = −0.264 [−0.387, −0.138], *p* < 0.001) and of social anxiety on suppression via the serial mediators FPE and acceptance (double mediation; β = 0.132 [0.069, 0.206], *p* < 0.001). Additionally, FNE mediated the link between social anxiety and rumination (β = 0.234 [0.170, 0.297], *p* < 0.001). All indirect effects are reported in Table 5.

With respect to the control variables, other gender (vs. female gender) was negatively associated with FNE (β = −0.051, *p* = 0.041). We found no other associations between gender or age and any of the study variables. Depressive symptoms were positively related to suppression (β = 0.197, *p* < 0.001) and rumination (β = 0.623, *p* < 0.001), and negatively related to acceptance (β = −0.315, *p* < 0.001).

The final model was able to explain 30.3% of the variance of acceptance, 34.6% of the variance of suppression, and 63.4% of the variance of rumination. For FPE and FNE, 60.8% and 50.8% of the variance could be explained, respectively. When depressive symptoms were excluded from the model, 24.5% of the variance of acceptance, 32.6% of the variance of suppression, and 44.4% of the variance of rumination could be explained. The final model, with all significant path coefficients, is depicted in Figure 2.

### 3.4. Multiverse Analyses

Following the recommendations of Simmons and colleagues [94] on *p*-hacking and the considerations of Steegen and colleagues [95] on the concept of multiverse analyses, we calculated a set of additional models to ensure full transparency of our analyses and to test the robustness of our results. For this purpose, we re-computed a number of models using (1) the dataset that we utilized for our main analysis, (2) the full dataset without excluding any cases, and (3) a dataset that excluded all participants who completed the questionnaire in less than 10 min. With each of these datasets, we specified eight models: two models with a single indicator, with and without covariates (Models a and b), two models with all items as indicators, with and without covariates (Models c and d), two models with the five best indicators, with and without covariates (Models e and f), and two models with manifest variables, with and without covariates (Models g and h). This resulted in 24 models in total, ranging from Model 1a to Model 3h. Across all models, these supplementary analyses yielded no substantial deviations with respect to the results of our main hypotheses. Minor deviations from our main model were the following: First, the direct effect of social anxiety on suppression could not be found consistently across all additional models (see Appendix A). Second, social anxiety had a direct effect on rumination when depressive symptoms were omitted. Third, the control variables gender and age had significant effects on some of the endogenous variables in some models, albeit quite weakly. All models except the models that had all items as indicators (models likely to be underpowered) yielded acceptable model fits. Taken together, these additional analyses support the robustness of our findings and the viability of our hypotheses. Exemplarily, a coefficient plot for the indirect effects of the double mediation (SA → FPE → ACC → SUP) across all models is displayed in Figure 3. Full results of the complementary analyses and coefficient plots for the other significant indirect effects in the model can be found in the electronic supplement (see Appendix A).

## 4. Discussion

FPE has recently emerged alongside FNE as a core cognitive component of social anxiety. It is thought to be one of the driving forces behind the positivity impairment often experienced by socially anxious individuals, whereas FNE is associated with an excessive amount of negativity, for example, in the form of negative and repetitive automatic thoughts [23]. Given that social anxiety is associated with various deficits in emotion regulation, the purpose of this study was to examine the relationships between both types of fear of evaluation and three emotion regulation strategies, i.e., acceptance, suppression, and rumination, and to assess their potential mediating role in the established links between social anxiety and emotion dysregulation in adolescents.

The analyses provided support for some, but not all, of our hypotheses. With respect to our first hypothesis, we found that after controlling for the unique contribution of social anxiety, the two types of fear of evaluation were associated with different emotion regulation strategies. Consistent with our expectations, FNE was associated with rumination. However, we did not find an association between FNE and either acceptance or suppression. Conversely, we were able to demonstrate an association between FPE and acceptance but not with suppression or rumination. Interestingly, social anxiety was associated with acceptance and rumination only in bivariate correlations but not in our higher-order correlational model (i.e., when the evaluation fears were considered simultaneously). In contrast, FPE and FNE appeared to be strongly associated with these two variables, above and beyond social anxiety. These initial findings already suggest that these two types of fear of evaluation are not only distinct constructs, as the extant literature suggests [26], but also have different implications for the way affected individuals manage their emotions.

While the bivariate correlations revealed a *positive* association between social anxiety and suppression, social anxiety was *negatively* associated with suppression in the final model. One explanation of this (at first sight) counterintuitive finding may be that social anxiety acts as a suppressor variable in this model, which requires further investigation. After examining the bivariate correlations of all our study variables, we found that social anxiety was strongly correlated with rumination, fears of evaluation, depression, and acceptance (see Table 2), whereas it was only moderately correlated with suppression. This could be the reason why social anxiety seems to act as a suppressor variable. We tested this by conducting hierarchical regression analyses in which suppression was the dependent variable and FPE, FNE, social anxiety, and depression were the independent variables. We found that the inclusion of social anxiety in the last step of the model indeed led to an increase in the standardized regression coefficients of FPE and FNE. Since suppression effects can be considered a function of multicollinearity [96], we re-examined the tolerance and variance inflation factors to test whether the inclusion of social anxiety in the model was justifiable. We found that, in fact, neither tolerance nor VIF of social anxiety exceeded the commonly used thresholds, even the more conservative ones [96] (Tolerance > 0.25, VIF < 4). We also computed a model without social anxiety as a predictor and examined the effects of FPE and FNE on acceptance, suppression, and rumination (see Appendix A). We found that the coefficients remained significant and were not inflated compared to the main model and, therefore, decided to include social anxiety in the final model.

That said, theoretical considerations could also explain this particular result. Watson et al. [86] and Paulhus et al. [97,98] have demonstrated that negative suppressor variables can indeed sometimes be theoretically relevant and should be carefully examined. In our models, we observed that the direct paths of social anxiety on suppression were either significant and negative or insignificant in many of our models. This would suggest that, if fears of evaluation and depressive symptoms are controlled for, social anxiety may no longer be significantly associated with suppression or may even be negatively associated with it. Although this finding appears counterintuitive, there are some studies supporting this notion. Firstly, Hofmann and colleagues [99], when validating their proposed scale to assess interpersonal emotion regulation, found that social anxiety was, in fact, positively correlated with most of the subscales of their instrument. For instance, individuals who reported fears of criticism and embarrassment, among other social anxiety-related measures, also appeared to endorse turning to others when they had negative emotions (example item: “When I feel sad, I seek out others for consolation”). In addition, Jose and colleagues [100] were able to demonstrate in a community sample of adolescents that social anxiety could predict co-rumination, which is defined as a tendency to extensively discuss personal problems within dyadic relationships. Although the association between social anxiety and suppression is very well established in the literature, these findings indicate that socially anxious individuals may tend to express and talk more about their emotions as long as they are not in an unfamiliar social situation in which fears of evaluation, positive or negative, take effect. On this note, given that the measure that we used in this study reflects a general tendency to suppress emotional reactions regardless of the context in which they are elicited, we consider it imperative to differentiate between various social situations in future studies. This would allow for conclusions to be drawn about the situations in which socially anxious individuals are more likely to suppress their emotions.

Regarding our mediation hypotheses, we found an indirect effect of social anxiety on acceptance via FPE but not FNE and an indirect effect of social anxiety on rumination via FNE but not FPE. Neither FPE nor FNE mediated the relationship between social anxiety and suppression. However, our analyses revealed a double mediation, i.e., an indirect effect of social anxiety on suppression via the mediators FPE and acceptance. FNE did not seem to play a role in the relationship between social anxiety, acceptance, and suppression.

Numerous studies have found that social anxiety is associated with the non-acceptance of emotions [58,59,60] as well as the suppression of emotions [61,62]. Other studies have consistently shown that a non-accepting attitude towards emotional experiences is associated with more negative and less positive affect [43]. On a similar note, suppression of emotions can have a counterproductive effect with regard to the subjective experience of emotions, i.e., negative emotions are amplified and positive emotions are dampened when suppressed [35]. These regulatory difficulties can be understood as aspects of the positivity impairment frequently observed in socially anxious individuals. FPE is thought to be of particular relevance to this positivity impairment, as it is associated with increased negative and decreased positive affect at both the state and trait levels [24] as well as the disqualification of positive social outcomes [26]. Our analyses provide additional support for this idea by showing that adolescents who reported high levels of FPE also reported difficulties in accepting their emotions, which, in turn, is associated with more frequent suppression of emotions. Thus, our findings suggest that it is via FPE that social anxiety is linked to non-acceptance and suppression. Additionally, consistent with Weeks and Howell’s thoughts on this matter [23], FNE could not explain these associations, which suggests that it is indeed mainly FPE that drives these positivity-related difficulties. Since the measure we used does not distinguish between the acceptance and suppression of positive or negative emotions butmeasures the general tendency to reject and suppress emotions of any valence, it is difficult to draw conclusions about the relationship between FPE and the emotion regulation of positive or negative emotions. Nonetheless, our analyses suggest that it is FPE rather than FNE that explains the apparent difficulty in accepting emotional experiences and, subsequently, suppressing them.

On the other hand, FPE does not appear to play a role in rumination, a well-established emotion regulatory deficit among socially anxious adolescents [61,63,64,65,66]. Socially anxious individuals have been shown to repeatedly harbor negative thoughts before or after social or performance situations, often focusing on the perceived negative aspects of said situations and brooding over them [51]. These findings suggest that FNE may be the underlying driver of the social anxiety–rumination link, as the negative aspects of self and the situation seem to be in focus. Hence, our finding is consistent with the notion that FNE is more specifically associated with excessive negativity in the context of social anxiety [23].

### 4.1. Practical Implications

SAD in children, adolescents, and adults has been shown to be relatively resistant to state-of-the-art treatment approaches, such as cognitive–behavioral therapy, compared to other anxiety disorders [101,102]. There are several explanations for this phenomenon, including a compromised ability to build stable and strong therapeutic relationships, difficulty performing exposure tasks due to impaired interpersonal interactions, and a general inability to achieve disconfirmation of fear expectations in social situations [102]. As research on FPE has proliferated over the past decade, it too has been identified as a potential barrier to achieving positive outcomes in CBT with socially anxious individuals. Given that most evidence-based treatment models, including CBT, focus on restructuring patients’ dysfunctional cognitions related to negative evaluation and that the mechanisms of change of many commonly used interventions depend on the individual’s ability to accept and integrate positive reinforcement into their life, fear and rejection of positive feedback might hinder a successful treatment process [20,77].

Our findings support this assumption by showing that FPE may not only act as a deterrent to positive reinforcement but may also have negative effects on individuals’ emotion regulation. Specifically, it appears to be associated with difficulties in the acceptance and subsequent expression of emotions, which, in a therapeutic setting, would compromise the utility of almost any targeted intervention for individuals. In addition, as considered by Hudson and colleagues [102], it could be one of the factors that prevent socially anxious individuals from forming strong therapeutic relationships. Furthermore, it could also be a reason for the difficulty in achieving disconfirmation of fearful expectations in social situations, as positive outcomes are regularly disqualified and positive feelings are not accepted or expressed.

Weeks and colleagues [103] developed a brief cognitive–behavioral treatment protocol specifically targeted at FPE. It included psychoeducation related to FPE, cognitive restructuring of FPE-specific negative automatic thoughts, and implementation of in-session and in-vivo exposure tasks focused on either self-presentation or accepting and receiving compliments. The initial results seem quite promising, as not only did FPE scores decrease after treatment but so did the severity of social anxiety symptoms in general. Moreover, studies have shown that acceptance-based treatment modules, such as ACT (acceptance and commitment therapy) [104], are viable alternatives to CBT in terms of symptom reduction in SAD [105,106] and that interventions targeting emotion dysregulation in SAD also lead to positive outcomes [46]. Taken together, fear of evaluation and emotion dysregulation are aspects of social anxiety that are ideally targeted simultaneously in therapeutic interventions by focusing on restructuring the cognitive biases associated with compliments and criticism, elucidating their possible connections to maladaptive emotion regulation strategies and promoting non-judgmental, accepting attitudes and appropriate expression of emotional experiences. This has the potential to reduce experiential avoidance and improve the ability to savor positive effects in social situations, ultimately leading to more sustainable treatment successes in the long term. Considering that adolescence is a period that is characterized by increases in evaluative fears and changes in emotion regulation strategy use [10,107], targeting adolescents’ fears of evaluation, positive and negative, as well as expanding their emotion regulation repertoire might prove to be particularly effective in this developmental period.

### 4.2. Limitations

This study has some limitations that need to be addressed. First and foremost, analyses in this study were conducted with cross-sectional data that were collected online from a self-selected convenience sample. On that note, it is noteworthy that our recruiting strategy revolved mainly around reaching adolescents on social media. We must, therefore, acknowledge that we might have reached participants that spend a lot of time on digital devices and social media. Given that problematic smartphone use has been linked to fears of evaluation [108], this might have amplified the aforementioned self-selection effect.

With regards to the cross-sectional structure of our data, since temporal ordering and elimination of competing explanations are, by definition, absent from our design, we cannot draw any conclusions about causal relationships among study variables from our analyses [109]. Thus, longitudinal designs, as well as experimental designs, are needed in order to accurately assess the proposed mediation effects. Additionally, the exclusive reliance on adolescents’ self-reports limits the implications of our results, given the neglect of others’ perceptions (e.g., parents, peers, clinicians) and the risk of inflated effect sizes due to common method variance. Furthermore, our sample was comprised primarily of individuals that self-identified as female, making generalizations to the broader (gender-diverse) adolescent population questionable.

Another important limitation of this study concerns our operationalization of emotion regulation. Although trait or habitual emotion regulation has been extensively studied in studies with cross-sectional designs, recent considerations have improved our understanding of the construct and have shed light on problems that might arise from such an approach [110]. Emotion regulation is a highly complex and dynamic process, which means that a pure black-and-white distinction between maladaptive and adaptive strategies most likely does not encompass all facets of functional emotion regulation. In fact, it is not possible to make any assumptions about adaptive or successful emotion regulation without taking into account situational-contextual and personality-related aspects [111]. It is, therefore, advisable to assess the flexibility of emotion regulation, i.e., state emotion regulation across different situations, for example, via ecological momentary assessment (EMA) [110], when trying to understand the role it plays in everyday life in different psychopathologies.

Furthermore, although screening instruments do not allow conclusions to be drawn about the presence of a mental disorder, it is noteworthy that 61.8% and 72.7% of our adolescent sample exceeded the commonly used cut-off values for depressive disorders (cut-off = 11 in the PHQ-9) and social anxiety disorders (cut-off = 24 in the SPIN) [112,113], indicating that the present sample is a particularly distressed group of adolescents. In fact, 25.5% of participants were in psychotherapeutic or psychiatric treatment at the time of survey response, which is well above average prevalence rates in the German-speaking area [114]. Although we cannot rule out the possibility that survey-specific characteristics contributed to these high numbers—such as the fact that the participants were a self-selected convenience sample whose underlying distress might have driven their participation, as well as the fact that this study was conducted during the COVID-19 pandemic, during which the prevalence rates of distress had risen significantly [115]—these are very high percentages that merit attention and must be taken into account when interpreting our results.

In this context, the fact that this study was conducted during the COVID-19 pandemic warrants discussion. Several pandemic-related items (concerning lockdown, schooling, quarantine, parental home-office, and other restrictions) were created and included in the first section of the online survey to assess participants’ circumstances at the time they completed it. Additionally, we assessed the overall subjective burden of the pandemic on a visual analog scale (VAS), ranging from 0–100. Bivariate correlations revealed that the VAS score was significantly associated with fears of positive and negative evaluation, social anxiety, rumination, and depressive symptoms after applying a Bonferroni correction, albeit relatively weakly (see Appendix A). However, since we controlled for depressive symptoms in all of our models, we chose to omit this variable from the main analyses. Nevertheless, the finding remains that adolescents who perceived the pandemic as more burdening also reported higher scores on the clinical scales as well as on rumination.

## 5. Conclusions

In this study, we examined the associations between adolescents’ social anxiety, fears of positive and negative evaluation, and three emotion regulation strategies, i.e., acceptance, suppression, and rumination. Additionally, we investigated the two types of fear of evaluation as potential mediators in the link between social anxiety and these emotion regulation strategies. Our results suggest that social anxiety is indirectly associated with suppression via FPE and acceptance and indirectly associated with rumination, mediated by FNE. These findings provide additional support that FPE might be the driving force behind the positivity impairment observed in socially anxious adolescents, whereas fear of negative evaluation is thought to be associated with an excess of negativity, for example, in the form of repetitive negative automatic thoughts. We eagerly await further research on the role of fears of evaluation in combination with emotion regulation in social anxiety in adolescence, specifically the fear of positive evaluation, which we consider a promising new avenue of research.

## Figures and Tables

**Figure 1 jcm-11-05979-f001:**
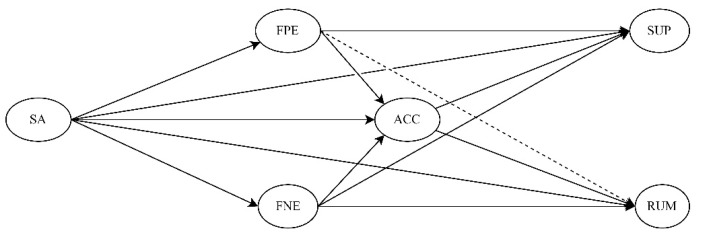
Conceptual model of fears of evaluation as mediators between social anxiety and emotion regulation. SA = social anxiety; FPE = fear of positive evaluation; FNE = fear of negative evaluation; SUP = suppression; ACC = acceptance; RUM = rumination. Control variables are not displayed for the sake of clarity. The dotted line represents the control path from FPE to RUM, assumed to be non-significant.

**Figure 2 jcm-11-05979-f002:**
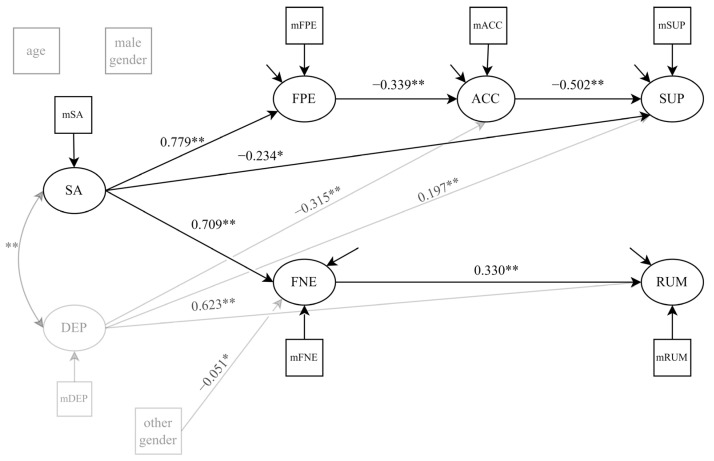
Structural equation model with standardized path coefficients. Control variables are displayed in grey. Non-significant paths are not shown. * *p* < 0.05; ** *p* < 0.01. SA = social anxiety; FPE = fear of positive evaluation; FNE = fear of negative evaluation; SUP = suppression; ACC = acceptance; RUM = rumination; DEP = depressive symptoms.

**Figure 3 jcm-11-05979-f003:**
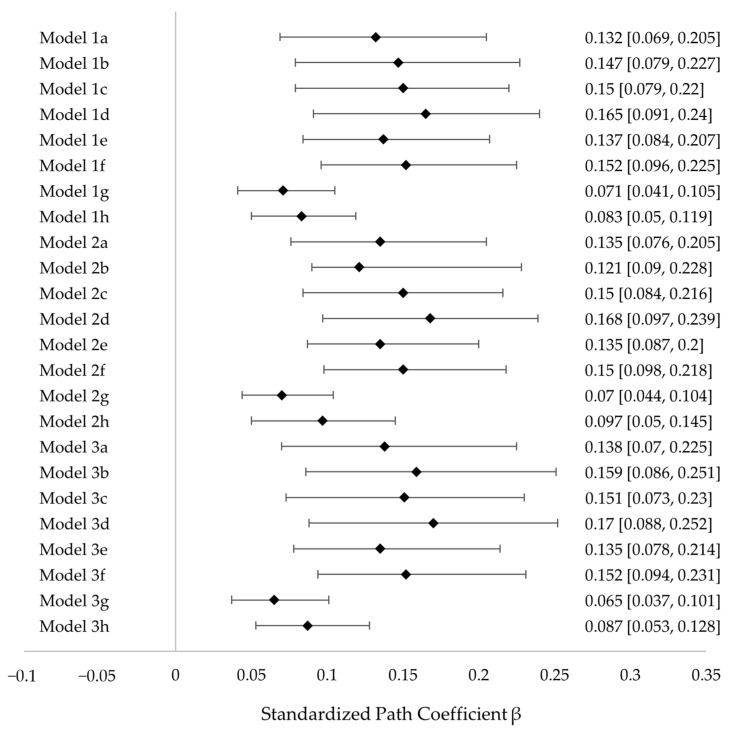
Coefficient plot displaying standardized coefficients of indirect paths from all calculated models. Indirect effect shown: social anxiety → FPE → acceptance → suppression.

**Table 1 jcm-11-05979-t001:** Sociodemographic variables.

Variable	*n*	%
*Gender*		
Male	65	10.0
Female	554	85.6
Other	28	4.3
*Residency*		
Austria	251	38.8
Germany	394	60.9
Italy	2	0.3
*First Language*		
German	579	89.5
Other	68	10.5
*Current Household (Living Together)*		
Both Parents	434	67.1
Mother	141	21.8
Father	22	3.4
Other	50	7.7
*Current Education*		
Middle School	4	0.6
High School	547	84.8
Vocational School	31	4.8
Special Needs School	6	0.9
Other	57	8.8
*Psychotherapy*		
Yes	165	25.5
No	482	74.5
*Physical Disability*		
Yes	61	9.4
No	586	90.6

Due to rounding inaccuracies, the percentages of male, female, and other gendered individuals add up to 99.9%.

**Table 2 jcm-11-05979-t002:** Means, standard deviations, and correlations.

Variable		*M*	*SD*	1	2	3	4	5	6
1.	SA	1.99	0.87	-					
2.	FNE	2.73	0.92	0.664 **	-				
3.	FPE	3.98	2.08	0.698 **	0.496 **	-			
4.	RUM	2.64	0.70	0.561 **	0.571 **	0.449 **	-		
5.	SUP	3.49	0.79	0.231 **	0.192 **	0.264 **	0.254 **	-	
6.	ACC	3.21	0.88	−0.389 **	−0.250 **	−0.409 **	−0.298 **	−0.471 **	-
7.	DEP	1.53	0.71	0.611 **	0.467 **	0.475 **	0.673 **	0.322 **	−0.404 **

*N* = 647. ** *p* < 0.01. SA = social anxiety; FNE = fear of negative evaluation; FPE = fear of positive evaluation; RUM = rumination; SUP = suppression; ACC = acceptance; DEP = depressiοn.

**Table 3 jcm-11-05979-t003:** Confirmatory factor analyses: model fit indices and composite reliabilities.

Questionnaire	RMSEA	CFI	TLI	SRMR	McDonald’s ω
SPIN	0.074 [0.067, 0.080]	0.920	0.905	0.050	0.929
FPES	0.066 [0.050, 0.083]	0.971	0.955	0.030	0.868
BFNE	0.070 [0.060, 0.079]	0.957	0.944	0.032	0.945
ASQ–Y: Acceptance	0.068 [0.060, 0.076]	0.922	0.902	0.063	0.835
ASQ–Y: Suppression					0.833
PTQ	0.066 [0.059, 0.074]	0.931	0.917	0.043	0.922
PHQ-9	0.073 [0.059, 0.086]	0.953	0.937	0.036	0.871

FPES = Fear of Positive Evaluation Scale; BFNE = Brief Fear of Negative Evaluation; ASQ–Y = Affective Style Questionnaire–Youth; PTQ = Perseverative Negative Thinking; PHQ-9 = Patient Health Questionnaire. One confirmatory factor analysis was conducted for both ASQ–Y scales.

**Table 4 jcm-11-05979-t004:** Direct effects of the structural equation model.

Direct Effects	*b*	BC 95% CI	β	BC 95% CI	*p*	*R* ^2^
*Fear of Positive Evaluation*						0.608
SA	**1.786**	[1.655, 1.906]	**0.779**	[0.732, 0.817]	<0.001	
Gender (F vs. M)	0.100	[−0.244, 0.468]	0.015	[−0.037, 0.070]	0.580	
Gender (F vs. O)	−0.016	[−0.512, 0.445]	−0.002	[−0.052, 0.046]	0.948	
Age	0.082	[−0.040, 0.205]	0.039	[−0.019, 0.097]	0.184	
*Fear of Negative Evaluation*						0.508
SA	**0.732**	[0.676, 0.794]	**0.709**	[0.667, 0.749]	<0.001	
Gender (F vs. M)	-0.128	[−0.324, 0.060]	−0.043	[−0.108, 0.021]	0.193	
Gender (F vs. O)	−**0.222**	[−0.436, −0.001]	−**0.051**	[−0.099, 0.000]	0.048	
Age	0.019	[−0.035, 0.072]	0.021	[−0.038, 0.077]	0.479	
*Acceptance*						0.303
SA	−0.014	[−0.210, 0.168]	−0.015	[−0.220, 0.180]	0.880	
FPE	−**0.138**	[−0.199, −0.073]	−**0.339**	[−0.490, −0.176]	<0.001	
FNE	0.063	[−0.040, 0.170]	0.070	[−0.044, 0.188]	0.243	
Depression	−**0.373**	[−0.510, −0.234]	−**0.315**	[−0.429, −0.197]	<0.001	
Gender (F vs. M)	−0.101	[−0.298, 0.091]	−0.037	[−0.111, 0.034]	0.309	
Gender (F vs. O)	−0.180	[−0.450, 0.122]	−0.045	[−0.114, 0.030]	0.216	
Age	0.032	[−0.033, 0.098]	0.038	[−0.038, 0.116]	0.331	
*Suppression*						0.346
SA	−**0.198**	[−0.366, −0.031]	−**0.234**	[−0.429, −0.038]	0.018	
FPE	0.035	[−0.022, 0.095]	0.096	[−0.060, 0.258]	0.239	
FNE	0.071	[−0.030, 0.172]	0.087	[−0.037, 0.210]	0.166	
Acceptance	−**0.454**	[−0.549, −0.359]	−**0.502**	[−0.599, −0.402]	<0.001	
Depression	**0.211**	[0.076, 0.347]	**0.197**	[0.071, 0.325]	0.002	
Gender (F vs. M)	−0.084	[−0.270, 0.107]	−0.034	[−0.113, 0.044]	0.379	
Gender (F vs. O)	0.142	[−0.126, 0.392]	0.040	[−0.034, 0.110]	0.280	
Age	−0.025	[−0.078, 0.026]	−0.033	[−0.101, 0.034]	0.336	
*Rumination*						0.634
SA	−0.065	[−0.177, 0.051]	−0.082	[−0.221, 0.064]	0.259	
FPE	0.023	[−0.017, 0.064]	0.066	[−0.049, 0.185]	0.268	
FNE	**0.254**	[0.186, 0.321]	**0.330**	[0.242, 0.417]	<0.001	
Acceptance	0.034	[−0.034, 0.105]	0.040	[−0.039, 0.122]	0.331	
Depression	**0.629**	[0.528, 0.726]	**0.623**	[0.531, 0.706]	<0.001	
Gender (F vs. M)	−0.047	[−0.182, 0.080]	-0.021	[−0.080, 0.034]	0.481	
Gender (F vs. O)	0.016	[−0.193, 0.236]	0.005	[−0.055, 0.068]	0.882	
Age	0.036	[−0.002, 0.076]	0.050	[−0.003, 0.107]	0.079	

*b* = unstandardized coefficient; β = standardized coefficient; SA = social anxiety; FPE = fear of positive evaluation; FNE = fear of negative evaluation; SUP = suppression; ACC = acceptance; RUM = rumination; DEP = depressive symptoms. Significant values are in bold.

**Table 5 jcm-11-05979-t005:** Indirect effects of the structural equation model.

Total and Indirect Effects	*b*	BC 95% CI	β	BC 95% CI	*p*
*Acceptance*					
Total Effect	−**0.214**	[−0.323, −0.100]	−**0.230**	[−0.345, −0.107]	<0.001
Total Indirect Effect	−**0.200**	[−0.344, −0.054]	−**0.214**	[−0.370, −0.058]	0.006
1. SA → FPE → ACC	−**0.246**	[−0.361, −0.130]	−**0.264**	[−0.387, −0.138]	<0.001
2. SA → FNE → ACC	0.046	[−0.029, 0.125]	0.050	[−0.032, 0.135]	0.245
*Suppression*					
Total Effect	0.015	[−0.101, 0.126]	0.018	[−0.121, 0.149]	0.798
Total Indirect Effect	**0.212**	[0.079, 0.346]	**0.252**	[0.095, 0.409]	0.002
1. SA → FPE → SUP	0.063	[−0.038, 0.171]	0.075	[−0.046, 0.202]	0.241
2. SA → FNE → SUP	0.052	[−0.021, 0.127]	0.062	[−0.026, 0.151]	0.168
3. SA → ACC → SUP	0.007	[−0.077, 0.097]	0.008	[−0.091, 0.114]	0.881
4. SA → FPE → ACC → SUP	**0.112**	[0.058, 0.176]	**0.132**	[0.069, 0.206]	<0.001
5. SA → FNE → ACC → SUP	−0.021	[−0.059, 0.013]	−0.025	[−0.070,0.014]	0.248
*Rumination*					
Total Effect	**0.155**	[0.082, 0.226]	**0.194**	[0.102, 0.284]	<0.001
Total Indirect	**0.220**	[0.133, 0.305]	**0.276**	[0.166, 0.379]	<0.001
1. SA → FPE → RUM	0.041	[−0.030, 0.115]	0.052	[−0.037, 0.145]	0.269
2. SA → FNE → RUM	**0.186**	[0.134, 0.239]	**0.234**	[0.170, 0.297]	<0.001
3. SA → ACC → RUM	0.000	[−0.015, 0.006]	−0.001	[−0.019, 0.008]	0.917
4. SA → FPE → ACC → RUM	−0.008	[−0.030, 0.007]	−0.011	[−0.038, 0.009]	0.360
5. SA → FNE → ACC → RUM	0.002	[−0.001, 0.011]	0.002	[−0.001, 0.014]	0.532

*b* = unstandardized coefficient; β = standardized coefficient; SA = social anxiety; FPE = fear of positive evaluation; FNE = fear of negative evaluation; SUP = suppression; ACC = acceptance; RUM = rumination; DEP = depressive symptoms. Significant values are in bold.

## Data Availability

All datasets, syntaxes, and statistical outputs of this article can be retrieved from OSF at https://osf.io/ceyws/ (accessed on 4 October 2022).

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
