# Peer review of "Please Don’t Compliment Me! Fear of Positive Evaluation and Emotion Regulation—Implications for Adolescents’ Social Anxiety"

_jcm, 2022, doi:10.3390/jcm11205979_

Round 1

Reviewer 1 Report

This paper is written very clearly and includes a well-ellaborated introduction section leading to the research questions and providing a profound rational for the SEM model tested The methodology used in this paper is well described and the authors have including several robustness checks of effects obtained. Moreover, implications of the present findings for the treatment of social anxiety disorder are well described in the discussion section. After having carefuly read the manuscript, I have some points / suggestions for further improving the paper, which I all regard as minor:

(1) @ Abstract: Instead of having a very trivial statement in the abstract ("Implications for the treatment of social anxiety are discussed") I would be nice to read 1 or 2 concrete implications for treatment in the abstract.

(2) @ Introduction 1.2 Emotion regulation: I'm wondering whether there is any literature on the rumination aspect with regard to positive events? I don't know any literature on that but I'm wondering what the authors' thoughts are. Regarding the aspect "unwillingness to accept negative emotions": I think, it would be worth to also discuss the aspect of "unwillingness to accept positive emotions" (~ disqualification of positive events) here. It is mentioned in the next chapter, but some thought about this at this point would be nice.

(3) @ Figure 1: Please add a footnote what the dotted line means (I think that no association is hypothesized between FPE and RUM).

(4) @ Recruitment strategy & limitations: I really acknowledge that the authors are transparant regarding the limitations of the study. However, I'm wondering what the authors' thoughts are about the limitation of the recruitment strategy (primarily social media) considering the topic of this paper (fear of negative/positive evaluation, social anxiety). Some reflections about this point in the limitation section would be beneficial.

(5) @ Participants, page 6, line 235: I think you have missed to mention the % of male participants in the text.

(6) @ Procedures: Please add the information whether the survey was anonymous or not.

(7) @ Measures: For the SPIN instrument a maximum score of 68 was mentioned. However, looking at Table 2 it seems that mean scores (and not sum scores) were used for the analysis (which seems to be true for all measures). If so, please briefly mention it in the text (and if so, the information of the maximum score for the SPIN instrument is not needed).

(8) @ Statistical analysis: I first thought that the information provided in the statistical analysis section of the paper is very brief. I then found much details about the analysis strategy in the results section which may be a bit uncommon. However, I think, this is indeed beneficial for understanding the results obtained. So, if the editor agrees with that, I'm also fine with reading details about the statisctial analysis in the results section and not the methods section.

(9) Results, page 11, line 109f: for consistency, please also use abbreviations for fear of positive / negative evaluation here.

(10) Results, Figure 3: I think, this Figure is not necessarily needed in the main part of manuscript. The authors may consider to move this figure to the supplementary material (but I would also agree to keep it in the manuscript).

(11) Discussion, page 17, line 649f: I would add that associations between pandemic burden and other outcome variables were rather weak.

Author Response

We used the template provided above. Please see the attachment.

Reviewer 2 Report

I have just one primary concern: the theoretical inconsistency between coping skills and emotion regulation strategies. Acceptance, suppression, and rumination are not emotion regulation strategies (see the five upper boxes in this Gross's emotion regulation model —https://psu.pb.unizin.org/psych425/chapter/process-model-of-emotion-regulation/). Coping is related to Lazarus's postulates (cognitive strategies to regulate emotional situations) [https://www.sciencedirect.com/topics/medicine-and-dentistry/coping-strategies]. Both topics are related but have different theoretical foundations. The authors used Gross's ER model but measured coping strategies instead of emotion regulation strategies.

Please fix this theoretical misunderstanding; the paper will be more theoretical sounding.

The rest of the paper is straightforward and matches the JARS (journal article research standards).

Best wishes

Author Response

(The authors gave the same response as above.)
